# Efficacy of Oncolytic Herpes Simplex Virus T-VEC Combined with BET Inhibitors as an Innovative Therapy Approach for NUT Carcinoma

**DOI:** 10.3390/cancers14112761

**Published:** 2022-06-02

**Authors:** Paul V. Ohnesorge, Susanne Berchtold, Julia Beil, Simone A. Haas, Irina Smirnow, Andrea Schenk, Christopher A. French, Nhi M. Luong, Yeying Huang, Birgit Fehrenbacher, Martin Schaller, Ulrich M. Lauer

**Affiliations:** 1Department of Medical Oncology and Pneumology, Virotherapy Center Tübingen (VCT), Medical University Hospital, 72076 Tübingen, Germany; vincent.ohnesorge@student.uni-tuebingen.de (P.V.O.); susanne.berchtold@uni-tuebingen.de (S.B.); julia.beil@klinikum.uni-tuebingen.de (J.B.); haass@biochem.mpg.de (S.A.H.); irina.smirnow@klinikum.uni-tuebingen.de (I.S.); andrea.schenk@med.uni-tuebingen.de (A.S.); 2German Cancer Consortium (DKTK), German Cancer Research Center (DKFZ), 72076 Tübingen, Germany; 3Department of Molecular Medicine, Max-Planck-Institute of Biochemistry, 82152 Martinsried, Germany; 4Department of Pathology, Brigham and Women’s Hospital, Harvard Medical School, Boston, MA 02115, USA; cfrench@bwh.harvard.edu (C.A.F.); nhi.luong@novartis.com (N.M.L.); yhuang44@bwh.harvard.edu (Y.H.); 5Department of Dermatology, University Hospital, 72076 Tübingen, Germany; birgit.fehrenbacher@med.uni-tuebingen.de (B.F.); martin.schaller@med.uni-tuebingen.de (M.S.)

**Keywords:** virotherapy, talimogene laherparepvec, T-VEC, NUT carcinoma, BET inhibitors, combination therapy

## Abstract

**Simple Summary:**

Since T-VEC is already approved for treatment of melanoma, its promising efficacy shown here also for NUT carcinoma (NC) cell lines may create a rapid transition to individual treatments as well as clinical trials in NC patients. The idea of combining T-VEC immunotherapy with BET inhibitors is strengthened by the assumption that the initial rapid response of NC to BET inhibitor therapy and the additional direct tumor cell lysis triggered by virotherapeutics may be able to effectively stabilize or even shrink the tumor cell mass to bridge the time gap until the durable immune response, induced by immunovirotherapy, can lead to complete tumor remission. This would signify a real breakthrough for patients suffering from this extremely aggressive tumor, whose average survival time is currently in the range of only six months.

**Abstract:**

NUT carcinoma (NC) is an extremely aggressive tumor and current treatment regimens offer patients a median survival of six months only. This article reports on the first in vitro studies using immunovirotherapy as a promising therapy option for NC and its feasible combination with BET inhibitors (iBET). Using NC cell lines harboring the BRD4-NUT fusion protein, the cytotoxicity of oncolytic virus *talimogene laherparepvec* (T-VEC) and the iBET compounds BI894999 and GSK525762 were assessed in vitro in monotherapeutic and combinatorial approaches. Viral replication, marker gene expression, cell proliferation, and IFN-β dependence of T-VEC efficiency were monitored. T-VEC efficiently infected and replicated in NC cell lines and showed strong cytotoxic effects. This implication could be enhanced by iBET treatment following viral infection. Viral replication was not impaired by iBET treatment. In addition, it was shown that pretreatment of NC cells with IFN-β does impede the replication as well as the cytotoxicity of T-VEC. T-VEC was found to show great potential for patients suffering from NC. Of note, when applied in combination with iBETs, a reinforcing influence was observed, leading to an even stronger anti-tumor effect. These findings suggest combining virotherapy with diverse molecular therapeutics for the treatment of NC.

## 1. Introduction

NUT carcinoma (NC), formerly known as NUT midline carcinoma (NMC), is a rare but very aggressive tumor, genetically defined by a reciprocal translocation involving the *NUTM1* (formerly known as NUT (nuclear protein in testis)) gene [1]. *NUTM1* is a rather unstructured protein with an unknown function that can be linked to various fusion partners, including *BRD4* (70%), but also *BRD3* [2], *NSD3* [3], or *ZNF532* [4]. Both BRD4 and BRD3 belong to the BET (bromo-domain and extra-terminal motif) protein family, which plays a crucial role in altering the transcription rate of certain genes, thereby affecting cell growth, and also leading to increased transcription of pro-inflammatory genes. Fusion with *NUTM1* then results in the expression of fusion oncoproteins that interact with chromatin and induce abnormal histone acetylation, which in turn leads to an epigenetic blockade of cellular differentiation and uncontrolled cell growth [4,5]. Histologically, NC usually presents as a monomorphic squamous cell carcinoma, most commonly with focal squamous differentiation or abrupt keratinization [1,6]. The incidence of NC cannot be accurately stated since too many cases are not diagnosed correctly. A large screening of >14,000 solid tumor samples found nine cases with *NUTM1* rearrangement [7], suggesting that NC is often underdiagnosed, probably due to lack of awareness.

The current treatment approach is surgical resection before or after chemoradiation, resulting in a median survival of only about six months [8,9,10]. This highlights the need for more specialized therapy modalities for this aggressive disease.

Even application of highly specific inhibitors of BET proteins (iBETs) [11,12] that reversibly bind the bromodomains of Bromodomain and Extra-Terminal motif (BET) proteins BRD2, BRD3, BRD4, and BRDT, and thereby prevent interactions between BET proteins and acetylated histones and transcription factors, is followed quite early by the development of high-grade resistance, toxicities at high iBET dosages and a relapse of tumor disease [13,14,15].

Moreover, results of recent preclinical and clinical studies (PMID: 32328561, 33311588, 29733771) indicate that this tumor type requires a combination of different treatment approaches, also including novel ones, to counteract the highly aggressive growth of NUT carcinoma.

Oncolytic immunotherapy has already demonstrated its potential for cancer treatment in other tumor types, but has not been investigated in NUT carcinoma so far. Oncolytic viruses (OVs) are replication-competent viral vectors, which are capable of selectively infecting tumor cells, replicating massively within and destroying infected tumor cells, concomitantly releasing thousands of progeny virus particles [16]. Furthermore, this process, called oncolysis, leads to the release of tumor-specific antigens in a highly inflamed tumor microenvironment, thus enabling a profound systemic anti-tumoral immune reaction, which is considered the key success factor of this biological cancer therapy [17]. Selectivity of OVs for cancer cells is due to defects found in many cancers, such as the loss of tumor suppressors, defective apoptosis pathways, activation of oncogenic signaling pathways, and most importantly, the loss of antiviral defense mechanisms [18]. Hence, oncolytic immunovirotherapy represents a promising therapeutic approach and also a potential combination partner for BET inhibitors for the treatment of NC. The rapid response of NC to iBET therapy [19] and the additional direct tumor cell lysis triggered by OVs may be able to effectively shrink the tumor cell mass to bridge the time gap until the durable immune response induced by immunovirotherapy can lead to a complete remission.

In this preclinical study, the oncolytic herpes simplex virus type 1 (HSV-1) *talimogene laherparepvec* (T-VEC), already approved for the treatment of melanoma [20], was used. T-VEC harbors two copies of the human granulocyte-macrophage colony-stimulating factor (GM-CSF) gene, inserted into the *ICP34.5*/*ICP47*-deleted version of strain JS1, which may further enhance the virus-induced systemic anti-tumor immune response [21]. The iBET compounds BI894999 and GSK525762, both targeting all four human BET proteins [22,23], were selected as combination partners. Both BI894999 and GSK525762 have already been tested in clinical trials (NCT01587703; NCT02516553) [19].

Here, the purpose was to investigate for the first time whether immunovirotherapy with T-VEC could become a new treatment option for NC patients. Moreover, combinatorial treatment with well-characterized iBET compounds was performed, in order to exclude any possible mutual negative effect of both therapeutic modalities as a basic prerequisite for clinical application, but also to investigate a possible additive anti-tumor effect. In this line, parameters such as cytotoxicity, tumor cell proliferation, viral replication, and interferon dependence of the oncolytic effect were investigated in six human tumor cell lines harboring the BRD4-NUT fusion oncoprotein.

## 2. Materials and Methods

### 2.1. Cell Lines

A panel of six human NUT carcinoma (NC) cell lines, all harboring the *BRD4-NUTM1* fusion gene, were employed in this study. NC cell lines 143100 and HCC2429 were kindly provided by Dr. Xin Zhang from University Hospital Essen, Germany. NC cell lines 14169, 10-15, and JCM were a kind gift from Dr. Chris French, Boston, MD, USA. NC cell line Ty-82 was purchased from JRCB Cell Bank (No.: JCRB1330). Cell line authentication was performed on all NC cell lines by STR profiling at the German Collection of Microorganisms and Cell Cultures (DSMZ) in Braunschweig, Germany. Vero cells (African green monkey kidney cells) were obtained from the DSMZ (No.: ACC 33) and were used for virus titration only. All cell lines were cultured in DMEM supplemented with 10% FCS. All cell lines were tested negative for mycoplasma contamination prior to utilization (MycoTOOL PCR Mycoplasma Detection Kit, Roche, Mannheim, Germany).

### 2.2. Treatment with BET Inhibitors (iBETs)

BI894999 (Amredobresib) was kindly provided by Boehringer-Ingelheim RCV (Vienna, Austria); GSK525762 (Molibresib) was obtained from Seleckchem (Planegg, Germany). NC cells were seeded in 24-well plates and treated 24 h later with the respective iBET compound. For this purpose, medium was replaced with cell culture medium containing iBETs at desired concentrations, and cells were incubated until their respective readout.

### 2.3. Virus Infection

The oncolytic herpes simplex virus type-1 (HSV-1) construct T-VEC (*talimogene laherparepvec*) was kindly provided by Amgen Inc. (Thousand Oaks, CA, USA). NC cell lines were infected with T-VEC as described previously [24]. For MOCK treatment, only serum-free DMEM was added. For combinatorial treatment, the infection medium was replaced at 1 hpi (hour post infection) with cell culture medium containing iBETs at respective concentrations.

### 2.4. Sulforhodamine B Cell Viability Assay

The viability of the six human NC cell lines was measured 96 h after iBET treatment, T-VEC infection, or combinatorial treatment in 24-well plates by using Sulforhodamine B (SRB) cell viability assay as described previously [24].

### 2.5. Real-Time Cell Proliferation Assay

NC cells were seeded in E-96-well-plates and viral infection and/or iBET treatment started 24 h later. Real-time dynamic cell proliferation was monitored in 30 min intervals during the complete observation period of 120 h using the xCELLigence RTCA SP system (Roche Applied Science, Penzberg, Germany). Cell index values were calculated using the RTCA Software (1.0.0.0805; Roche Applied Science, Penzberg, Germany).

### 2.6. Virus Growth Curve

NC cells were seeded in 6-well plates and infected with T-VEC at suitable MOIs with and without additional iBET treatment. Viral replication was quantified by performing plaque assays at 1, 24, 48, 72, and 96 hpi, as described previously [24].

### 2.7. GM-CSF ELISA

NC cells were seeded in 24-well plates and pretreated with 2 ng/mL IFN-β for 16 h before infection with T-VEC at cell line-adjusted MOIs. At 16, 24, 72, and 96 hpi, supernatants were harvested for quantification of T-VEC-mediated GM-CSF expression by using MAX™ Deluxe Human GM-CSF ELISA Kit (BioLegend, San Diego, CA, USA) according to manufacturer’s instructions.

### 2.8. Transmission Electron Microscopy

HCC2429 NC cells were infected with T-VEC at MOI 0.0001 or 0.001, trypsinized and fixed in Karnovsky fixative at 48 hpi and stored at 4 °C. Cell pellets were embedded as described earlier [24] and cell blocks were cut using an ultramicrotome (Ultracut, Reichert, Vienna, Austria). Ultra-thin sections (30 nm) were mounted on copper grids (Science Services, Munich, Germany) and analyzed using a Zeiss LIBRA 120 transmission electron microscope (Carl Zeiss AG, Oberkochen, Germany) operating at 120 kV.

### 2.9. Immunofluorescence Staining and Confocal Microscopy

The 143100 NC cells were seeded in tissue culture dishes and pretreated with 2 ng/mL IFN-β for 16 h or left untreated as control, before they were infected with T-VEC at MOI 0.1 or left uninfected (MOCK). At 8 and 16 hpi, cells were harvested and frozen for cryosectioning. Fresh frozen 5 µm sections were fixed with periodate–lysine–paraformaldehyde. Sections were blocked using donkey serum and stained with a primary rabbit-anti-HSV1 antibody (Novusbio, Littleton, CO, USA). Bound antibody was visualized using donkey-anti-rabbit-Cy3 (Dianova, Hamburg, Germany). For nuclear staining, Yopro (Invitrogen, Waltham, MA, USA) was used. Sections were analyzed using an LSM 800 confocal laser scanning microscope (Carl Zeiss AG, Oberkochen, Germany) and processed with the software ZEN 2.3 (blue edition; Carl Zeiss AG, Oberkochen, Germany).

### 2.10. Statistical Analysis

Statistical analysis was performed with GraphPad Prism Version 9 (GraphPad Software Inc., San Diego, CA, USA). Reduction in cell mass between two treatment groups was analyzed by Brown-Forsythe and Welch ANOVA and Dunnett’s multiple comparison test. In experiments regarding IFN-β pretreatment, significance between two treatment groups was analyzed by one-way ANOVA and Šidák’s multiple comparison test. Four different *p* values were determined: *p* < 0.05 (*), *p* < 0.01 (**), *p* < 0.001 (***), *p* < 0.0001 (****).

## 3. Results

### 3.1. Oncolytic Virus T-VEC Efficiently Infects and Lyses NC Cells

The clinically licensed virotherapeutic compound T-VEC was employed to analyze for the first time the general susceptibility of human NC cell lines to virus-mediated oncolysis and to find out whether the replication cycle of T-VEC in NC cells is comparable to other already characterized tumor entities.

In a first approach, transmission electron microscopy (TEM) images were taken exemplarily from T-VEC-infected HCC2429 NC tumor cells at 48 h post infection (hpi) (Figure 1). Steps of the well-described life cycle of wild-type HSV (Figure 1A) could be visualized in T-VEC-infected HCC2429 NC cells (Figure 1B), as previously also shown for neuroendocrine tumor (NET) cell lines [24]. Shown are the transport of newly formed viral capsids through the perinuclear space (Figure 1C), wrapping of viral capsids into vesicles of the trans-Golgi network (TGN) for final maturation in the cytoplasm (Figure 1D), the onset of T-VEC-mediated lysis of the tumor cell, indicated by a defect in the cellular membrane (Figure 1E), release of a small portion of mature progeny viral particles by exocytosis (Figure 1F), and finally complete oncolysis, leading to the release of a large number of T-VEC progeny particles (Figure 1G).

To assess the virotherapeutic efficiency, six different *BRD4-NUTM1* cell lines were infected with T-VEC and the remaining NC tumor cells were measured at 96 hpi using SRB cell viability assay (Figure 2). All NC cell lines were found to be susceptible to virotherapy, even at very low concentrations of T-VEC. This feature of a very pronounced T-VEC mediated oncolytic effect is exemplified by NC cell lines 143100 (Figure 2A), HCC2429 (Figure 2B), and 10-15 (Figure 2D), which proved to be particularly sensitive with significant reductions in NC tumor cells at the lowest tested MOI of 0.0001. Higher MOIs (up to an MOI of one) led to a complete eradication of NC tumor cells (remnant viable NC cells < 10%) in all six NC cell lines.

### 3.2. Pretreatment with IFN-β Heavily Impairs the Oncolytic Capacity of T-VEC in NC Cells

Preliminary experiments have shown that none of the six NC cell lines intrinsically produce IFN-β or are able to react to infection with T-VEC by induction of an IFN-β response (data not shown). However, when NC tumor cells were pretreated with IFN-β for 16 h, SRB assays at 96 hpi revealed that the potent oncolytic activity of T-VEC was almost completely impaired when compared to the IFN-β-untreated control (Figure 3A and Appendix A), albeit depending on the NC cell line studied. Accordingly, in the NC cell lines 143100, HCC2429, Ty-82, 10-15, and JCM, an almost complete suppression of oncolytic efficiency could be induced, whereas in the NC cell line 14169, IFN-mediated blockage could be overcome to a minor extent with increasing virus concentrations.

Moreover, quantification of T-VEC-mediated GM-CSF expression in 143100 NC cells at different time points during the course of infection by ELISA indirectly provided insights into the timeline of impairment of T-VEC-mediated gene expression and thus viral replication of T-VEC (Figure 3B). While GM-CSF expression in IFN-untreated NC cells started as early as 16 hpi with the higher MOI of 0.001 and increased significantly by 24 hpi, expression of GM-CSF was found to be completely impaired in IFN-pretreated cells at these two early time points. At later timepoints, a sharp increase in GM-CSF expression was also observed exclusively in IFN-untreated NC cells until 72 hpi, after which a plateau was reached (Figure 3B).

Similar results were observed for the other five NC cell lines (Appendix A). In NC cell lines HCC2429, Ty-82, and 10-15, IFN-β pretreatment resulted in suppression of GM-CSF expression at both examined time points (72 and 96 hpi). Solely in the NC cell line 14169, these observations could not be reproduced as clearly. Although GM-CSF expression was significantly reduced in IFN-pretreated compared to -untreated cells both at 72 and 96 hpi, a complete prevention of T-VEC marker gene expression was not seen in this cell line (Appendix A). In the NC cell line JCM, T-VEC replication was also not completely suppressed with the higher MOI tested; however, it was significantly reduced compared to the omission of IFN-β pretreatment (Appendix A).

Furthermore, early infection processes were visualized by immunofluorescence staining in both IFN-pretreated and -untreated 143100 cells after infection with a high T-VEC MOI of 0.1 (Figure 3C and Appendix A). In IFN-untreated cells, initial T-VEC infection could be shown as early as 8 hpi, which is restricted exclusively to the nucleus at this early time point. In comparison, at 16 hpi virus particles could also be detected in the cytoplasm of the cells, indicating a later step in the life cycle of HSV-1 viruses, namely the packaging and transport of a large number of progeny virus particles to the outer cell membrane (Figure 1D). In contrast, in IFN-pretreated cells, the onset of infection could be detected by T-VEC staining exclusively in the nucleus at 16 hpi (Figure 3C and Appendix A).

### 3.3. iBET Therapy in Addition to T-VEC Therapy Leads to an Enhanced Anti-Tumoral Effect

In order to develop a potent and interacting therapeutic approach for highly aggressive NC tumors, a combination of oncolytic immunovirotherapy with iBETs was investigated next. First, all six NC tumor cell lines were treated with the iBET compounds BI894999 or GSK525762 in monotherapy. Here, the anti-tumoral effects of both compounds that are already under clinical investigation for NC treatment could be confirmed for all investigated NC cell lines, using SRB cell viability assay (Appendix A). An iBET concentration-dependent decrease in tumor cells could be observed in every NC cell line with a range in concentrations from 0.5 to 5 nM for BI894999 and 0.1 to 10 µM for GSK525762.

In order to find out whether or not T-VEC could become a suitable partner for a combinatorial therapy with iBET compounds, SRB cell viability assays were performed in the six NC cell lines after combinatorial treatment (Figure 4). T-VEC MOIs and iBET concentrations were adjusted so that a residual NC tumor cell mass of approximately 75% was achieved when applying the respective monotherapeutic approaches. Thus, in subsequent combinatorial testing, both inhibitory and additive interactions of both drug classes (OV/iBET) could be observed. Four out of six NC cell lines (HCC2429, Ty-82, 14169, and JCM) displayed a significantly enhanced anti-tumoral effect in the combinatorial approach compared to the most effective monotherapy (*p* < 0.01 to *p* < 0.0001), respectively. While for the NC cell line 10-15, neither T-VEC alone nor the combinatorial approach was able to outperform the iBET monotherapy (Figure 4D); for the cell line 143100, it was difficult to monitor additional enhancing effects, due to the highly potent T-VEC therapy (Figure 4A).

In order to verify these results, real-time cell proliferation was monitored over 120 h in all six NC cell lines treated either with T-VEC or iBETs as monotherapy in cell line-adapted concentrations or in a combinatorial setting (Figure 5 and Appendix A). The results of the SRB assays could be confirmed in the real-time cell proliferation studies. In detail, the combination of iBET and T-VEC additionally increased cytotoxicity in 143100 NC tumor cells compared to each monotherapeutic use beginning at 48 hpi and led to a total stop of cell proliferation at 72 hpi (Figure 5). When comparing all six NC cell lines, two different early response patterns to iBET monotherapy were noted (Appendix A): In three NC cell lines (143100, HCC2429, and 14169) an increase in cell proliferation started around 12 hpi and lasted until 48 hpi. In the other three NC cell lines (Ty-82, 10-15, and JCM), this increase was not noticed. A possible explanation for this apparent increase might be the induction of cellular senescence, leading to a change in morphology, which was found to be induced in mucoepidermoid carcinoma (MEC) cell lines after treatment with the iBET GSK525762 [25]. These experiments show that although iBET monotherapy led in some NC cell lines (Appendix A for BI894999; Appendix A for GSK525762) to decreased cell proliferation, it did not induce cell death because the cell index curve only flattened, but did not drop. However, this decline, indicative of cell death, was seen with both T-VEC monotherapy and combinatorial approaches, but with an onset 12 h earlier with combinatorial treatment than with monotherapy.

### 3.4. iBET Therapy Does Not Impair Viral Replication of T-VEC in NC Tumor Cells

Viral titers of T-VEC were determined at a low MOI of 0.0001 over 96 h in the absence and presence of the two iBET compounds in all six NC cell lines to ensure that infection and replication of T-VEC in NC cells were not negatively affected by subsequent iBET therapy, despite visible additive effects of both agents (Figure 6).

As expected, viral titers increased substantially, with a more rapid time to plateau in NC cell lines with higher susceptibility to the virus. In general, the maximum viral titers that were reached in cells with T-VEC monotherapy were between 10^6^ and 10^7^ plaque-forming units (PFU)/mL for all tested NC cell lines, indicating a high susceptibility of all six NC cell lines to T-VEC infection. Thereby, a correlation between the reduction in viable tumor cells in the SRB assay and the viral titers could be observed: in the more susceptible NC cell lines 143100, HCC2429, and 14169, viral titers increased up to 10^7^ PFU/mL (Figure 6A,B,E). On the one hand, in NC cell lines 143100 and 14169, a plateau was already reached at 48 hpi, and even a small decrease was observed at 96 hpi, indicating strong oncolysis. On the other hand, the less susceptible NC cell lines Ty-82, 10-15, and JCM were found to exhibit lower viral titers (up to 10^6^ PFU/mL) without a decrease at 96 hpi (Figure 6C,D,F).

Importantly, titers of T-VEC in the combinatorial approaches were at least as high as without the additional iBET treatment. For NC cell line 10-15, even a significant induction of the viral growth could be observed after 72 h.

## 4. Discussion

Immunovirotherapy is a new option for the treatment of NC patients as its approach may help to overcome the limitations of other therapeutic approaches [26]: (i) surgical resection often is impossible due to an early presence and rapid spread of metastases; (ii) during (neo-) adjuvant chemoradiation, tumor cells are able to escape apoptosis; and (iii) any initial response to BET inhibitors is followed early by the development of high-grade resistance [13,14,15]. With the rapid growth of NC, few surviving tumor cells often appear to be sufficient for tumor recurrence, as evidenced by the high progression rates after initial radical therapies and only a few long-term survivors [8,9,10].

Notably, oncolytic viruses are able to indirectly target these small tumor remnants or metastases, as they can activate the body’s immune system to provide an in situ anti-tumoral immune response [27], and are therefore promising candidates for therapeutic combination.

In this study, six human NC cell lines harboring the BRD4-NUT fusion oncoprotein were evaluated for their response to immunotherapy with T-VEC, either alone or in combination with one of two different BET inhibitors (BI894999; GSK525762).

As a major result, it was shown that in all six NC cell lines, monotherapeutic treatment with T-VEC resulted in a significant reduction in tumor cells when applied at sufficiently high MOIs. However, in order to better classify the virus concentrations used, a systematic review of the efficacy of this immunotherapy in a variety of different tumor cell lines, such as those included in the NCI-60 tumor cell panel, would be beneficial. Such systematic NCI-60 screens have already been performed with other virotherapeutics, for example, with an oncolytic measles vaccine virus [28] or with an oncolytic vaccinia virus [29], in which the 54 adherently growing tumor cell lines of the NCI-60 panel were tested for their sensitivity to infection. Thereby, tumor cell lines were classified as susceptible to the tested immunotherapeutic agent if treatment with a defined MOI for 96 h resulted in a decrease in residual tumor cell masses of <50% compared to MOCK treatment, defined as partially resistant with a residual cell mass between 50% and 75%, and defined as resistant with a residual cell mass of >75%. Since T-VEC, in general, appears to be much more potent, detailed characterization of NC cell line-specific response rates was found to be possible only at a very low MOI. For example, if an MOI of 0.0001 would be set as the standard in the NC tumor setting here, the following pattern of differences would result for the six NC cell lines examined: (i) cell lines 143100 and HCC2429 would be highly susceptible; (ii) cell lines 10-15 and 14169 would be classified as partially resistant; and (iii) Ty-82 and JCM would be declared as resistant to infection with T-VEC. However, due to this very low viral concentration, greater biological variation may occur in the performed cell culture assays, explaining larger standard deviations (for example in the SRB assays in Figure 2).

Type I interferons, including IFN-β, play an essential role in the immune response of healthy cells to viral infections as they can diminish the replication of viral genomes, protein translation, and virus egress [30]. In cancer cells, these interferon signaling pathways are frequently disrupted, allowing unlimited replication and thus increasing the efficiency of viral oncolysis [31], which is a prerequisite for the success of oncolytic virotherapy.

In order to demonstrate the general sensitivity of T-VEC to IFN-β and conversely show that NC tumor cells may be particularly amenable to immunovirotherapy with T-VEC due to possible defects in IFN signaling pathways, NC tumor cells were pretreated with IFN-β, before infection with T-VEC, and subsequently, anti-tumor efficacy was investigated. As a result, it was demonstrated that in all examined NC cell lines, pretreatment with IFN-β resulted in massively reduced oncolytic efficiencies, which was conversely confirmed by quantification of T-VEC-mediated GM-CSF expression. Of note, the sensitivity of T-VEC to IFN-β was found to vary depending on the NC cell line studied and the MOI used, indicating that there may be an NC cell line-specific critical virus concentration at which IFN-triggered inhibition can be overcome.

The observation that pretreatment with IFN-β suppresses T-VEC-mediated gene expression at very early time points after T-VEC infection (at 16 hpi) suggests that not only viral replication but also possibly initial infection of NC tumor cells may be suppressed. However, early infection processes visualized by immunofluorescence staining in both IFN-pretreated and untreated 143100 NC cells provide evidence that IFN, at least in this NC cell line, is unable to completely prevent T-VEC infection induced by a quite high viral concentration (MOI 0.1), but nevertheless slows down the replication process significantly. Both the results obtained here, and the observation that none of the six NC cell lines intrinsically produce IFN-β or are able to react to T-VEC infection with an IFN-β response (data not shown), suggest that NC tumors indeed are particularly well suited for immunotherapy with this generally IFN-sensitive virotherapeutic agent and that a possible explanation for this may lie in distinct defects in IFN signaling pathways in this hitherto rather uncharacterized tumor entity. Whether this assumption can be confirmed must be clarified in further investigations.

In many clinical studies, carried out with different iBET agents, NC tumors initially responded well, but progression often started again only after weeks or months [19,32,33]. These first studies indicate that (neo-)adjuvant iBET treatment might be a suitable treatment option for NC patients before or after surgical resection, but it fails to offer a sufficient and long-lasting therapy option for NC patients in the context of an iBET monotherapy [34]. Therefore, the search for suitable combination partners for iBET therapy was initiated, focusing mainly on other small molecules, with a few preclinical studies testing the PD-1/PD-L1 blockade as a representative of immunotherapy with the first promising results [22,34,35].

A combination of T-VEC-mediated immunotherapy with iBET compounds provides the opportunity to initially shrink the NC tumor cell mass of this fast-growing and aggressive tumor through a direct reinforcing attack consisting of the antiproliferative iBET therapy and the oncolytic effect of T-VEC, thus keeping the tumor under control until the long-lasting anti-tumor immune response triggered by T-VEC becomes effective and can lead—in the best case—to a complete tumor remission.

In our preclinical study presented here, the groundwork for a combination therapy was laid for the first time by ensuring in vitro that the two therapeutic approaches (OV/iBET) do not generally affect each other negatively. Combinatorial treatment of the six NC cell lines with T-VEC and iBET compounds BI894999 or GSK525762 showed no evidence of mutual negative interference, either in terms of cell mass reduction or viral replication. Furthermore, in five out of six cell lines a significantly enhanced NC tumor cell mass decrease could be observed compared to the corresponding most effective monotherapy. A recent study using several iBETs, including NEO2734, GNE-781, and OTX015, to treat NC cell lines demonstrated that these iBETs are able to suppress the IFN response in NC cells [36,37], which hypothetically could enhance the effect of oncolytic viruses in general because of their sensitivity to IFN. Whether this effect is also responsible for the observed enhanced decrease in NC cell masses in our combination approach needs to be clarified in further studies. In addition, it would have been interesting to further investigate specific cellular immune responses towards combinatorially treated (T-VEC/iBET) NC cells; however, to date, there are no adequate cell culture models that would allow such experiments.

In our combinatorial approaches, it was quite challenging to choose NC cell line-specific suitable MOIs and concentrations due to the high susceptibility of the NC cell lines to T-VEC as well as to the iBET compounds. The initial goal of achieving a viable NC tumor cell mass of approximately 75% with monotherapeutic treatment was met only with difficulty, as T-VEC proved to be exceedingly effective in terms of its virus-induced oncolysis of NC cells. Considering the possible biological variations at these low concentrations of T-VEC, this may have led to the observed non-significant reduction in 10-15 NC tumor cells after combination therapy. While this is a problem for in vitro tests, it could be an advantage for clinical applications. Thus, in initial clinical trials using GSK525762 for the treatment of NC, a dose of 80 mg/day was found to be well tolerated and resulted in plasma concentrations of 2.5 µM at baseline [19]. Therefore, the results can be considered to confirm the benefit of GSK525762 seen in vitro for NC treatment in the clinical setting.

In 2006, it was shown that BRD4 represses gene translation of the human papillomavirus [38]. It was hypothesized that a similar mechanism might be possible for T-VEC, resulting in induced viral replication after application of iBET and consequently further reduced tumor cells. However, this could not be confirmed when T-VEC replication was examined in combination with iBETs in general, as only an increased viral replication in NC tumor cell line 10-15 was detectable. Nonetheless, these results indicate that simultaneous application of both substance classes (OV/iBET) possibly can lead to an improved NC therapy.

In this study, only NC cell lines harboring the most often seen *BRD4-NUTM1* translocation were investigated. Stirnweiss et al. reported on a study comparing the general efficiency of iBET therapy in different NC cell lines. It was shown that the type of *NUTM1* fusion expressed may be one of the factors that contribute to iBET sensitivity, with NC tumor cells expressing the *BRD4-NUTM1 ex11:ex2* fusion being on average more than tenfold more sensitive to iBET treatment than cells with *BRD4-NUTM1 ex15:ex2* or *ex14:ex2* fusions [39]. Unfortunately, this cannot be compared with our data because other iBET compounds were used. At the current stage, it remains unclear whether the translocation type has any importance towards resistance phenomena to T-VEC-mediated immunovirotherapy, while differential responsiveness to iBET compounds due to different breakpoints seems likely. Therefore, further studies comparing genomically profiled NC tumor cell lines with *BRD3*, *NSD3,* or other gene fusion characteristics are needed.

## 5. Conclusions

Our data demonstrate that immunotherapy with T-VEC, even as monotherapy, constitutes a highly efficient tool for the treatment of NUT carcinoma cells. As a next step, it would be beneficial to test the efficacy of this therapeutic regimen in NC organoid models and further in immunocompetent mouse models. To date, only immunodeficient mouse models could have been developed [12,36,40,41], which are not suitable to study the immunologic anti-tumor processes triggered by immunovirotherapy.

In a recent phase Ib clinical trial undertaken in advanced melanoma patients, it was shown that T-VEC combined with anti-PD-1 heightens the treatment efficiency compared to treatment with anti-PD-1 alone, due to changes in the tumor microenvironment [42]. However, a subsequent phase III trial failed to demonstrate an additive effect of combining the two treatments in the first-line setting (NCT02263508) [43] but studies evaluating the combination in the neoadjuvant and post-PD-(L)1 setting (NCT04068181), and also in tumor types other than melanoma (NCT02509507), are still ongoing. Given that the PD-1/PD-L1 blockade has now also been successfully tested in NC patients [44], a triple combinatorial approach consisting of T-VEC, iBET compounds, and immune checkpoint inhibitors may further improve therapeutic efficacy. With T-VEC already clinically approved and both iBET compounds already under clinical investigation, the dual approach, as well as a possible triple approach with anti-PD-1/anti-PD-L1, offers the potential for rapid development from “bench to bedside”.

## Figures and Tables

**Figure 1 cancers-14-02761-f001:**
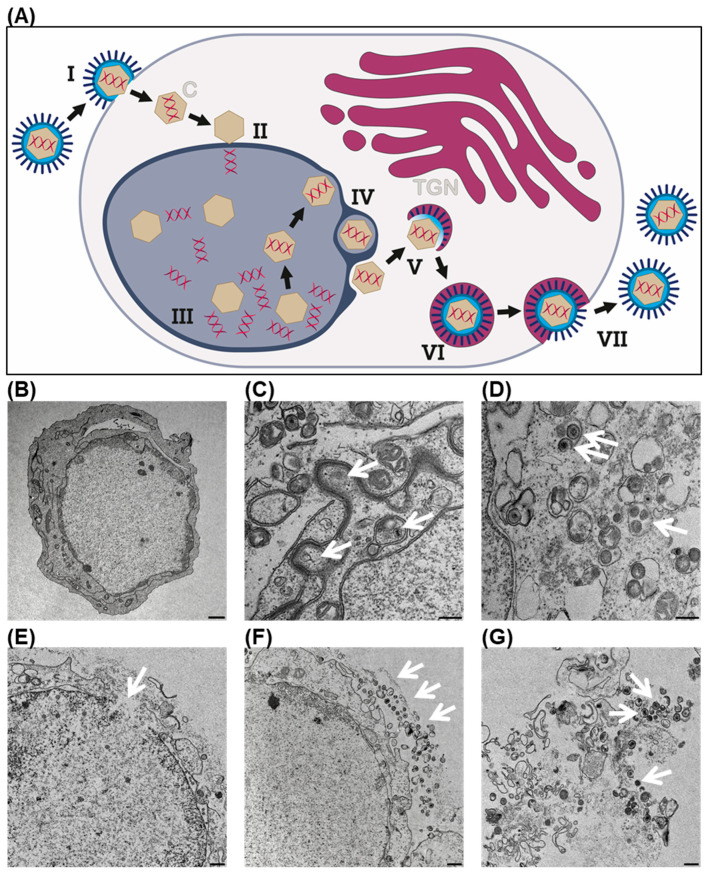
The life cycle of a herpes-simplex-virus type 1 (HSV-1)-based virotherapeutic (T-VEC) (**A**) and transmission electron microscopy (TEM) images of T-VEC-infected HCC2429 NUT carcinoma cells at 48 h post infection (hpi), illustrating individual steps of HSV-1 propagation (**B**–**G**): (**A**) (I) Glycoproteins located on the surface of HSV-1 virus particles attach to cellular receptors, followed by fusion of the viral envelope membrane with the cytoplasmic membrane of the target cell to release the virus capsid (**C**) into the cytoplasm. (II) The HSV-1 capsid migrates along the cytoskeleton to the cell nucleus, where it attaches; the viral DNA is released into the nucleus, leaving the empty capsid behind. (III) In the nucleus, transcription of viral genes and genome replication take place, thus enabling the assembly of progeny viral capsids. (IV) Newly formed viral capsids first attach to the nuclear lamina, are then transported through the inner nuclear membrane, and finally released into the cytoplasm. (V) Final maturation of the capsid occurs via budding into vesicles of the trans-Golgi network (TGN), which contain viral glycoproteins (dark blue spikes). (VI) Enveloped virions within cellular vesicles are transported to the cell surface. (VII) Then, vesicle and plasma membranes fuse in order to release a mature, enveloped progeny HSV-1 virion from the cell. (**B**) Overview of a single T-VEC-infected HCC2429 NUT carcinoma cell. Scale bar shows 1000 nm. (**C**) Transport of newly formed viral capsids (white arrows) through the perinuclear space (step IV in (**A**)). Scale bar shows 250 nm. (**D**) In the cytoplasm, viral capsids are wrapped into vesicles of the trans-Golgi network (TGN) for final maturation (white arrows; step V in (**A**)). Scale bar shows 250 nm. (**E**) Onset of oncolysis in a T-VEC-infected HCC2429 cell, indicated by a defect in the cellular membrane (white arrow). Scale bar shows 500 nm. (**F**) Multiple mature progeny T-VEC particles are released by exocytosis, lining still up on the outside of the cellular membrane (white arrows) (step VII in (**A**)). Scale bar shows 500 nm. (**G**) Completed oncolysis results in the release of a multitude of T-VEC progeny particles (white arrows). Scale bar shows 500 nm.

**Figure 2 cancers-14-02761-f002:**
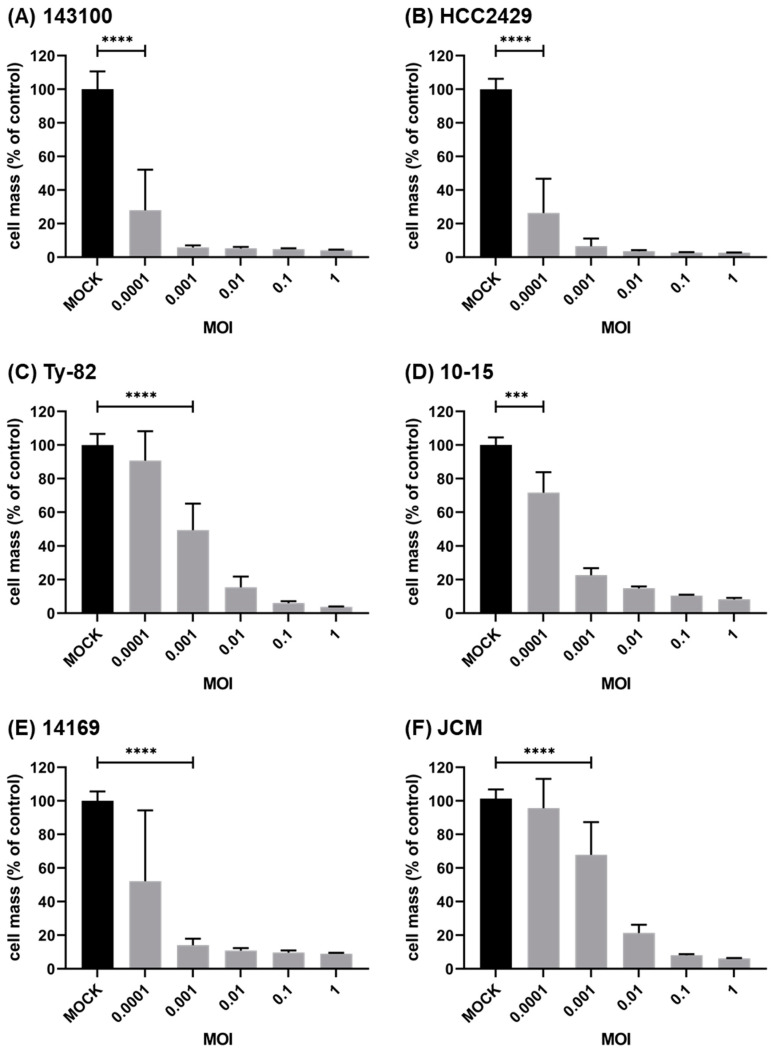
Viability of human NC cell lines after monotherapeutic treatment with T-VEC: 143100 (**A**), HCC2429 (**B**), Ty-82 (**C**), 10-15 (**D**), 14169 (**E**), and JCM (**F**) NC tumor cells were infected with T-VEC at different multiplicities of infection (MOIs) ranging from 0.0001 to 1 or remained uninfected (MOCK). At 96 h post infection (hpi), the remaining NC tumor cells were determined by SRB viability assay. T-VEC-mediated oncolysis was calculated relative to MOCK control. The mean ± SD of at least two independent experiments performed in quadruplicates is shown. *** *p* < 0.001, **** *p* < 0.0001.

**Figure 3 cancers-14-02761-f003:**
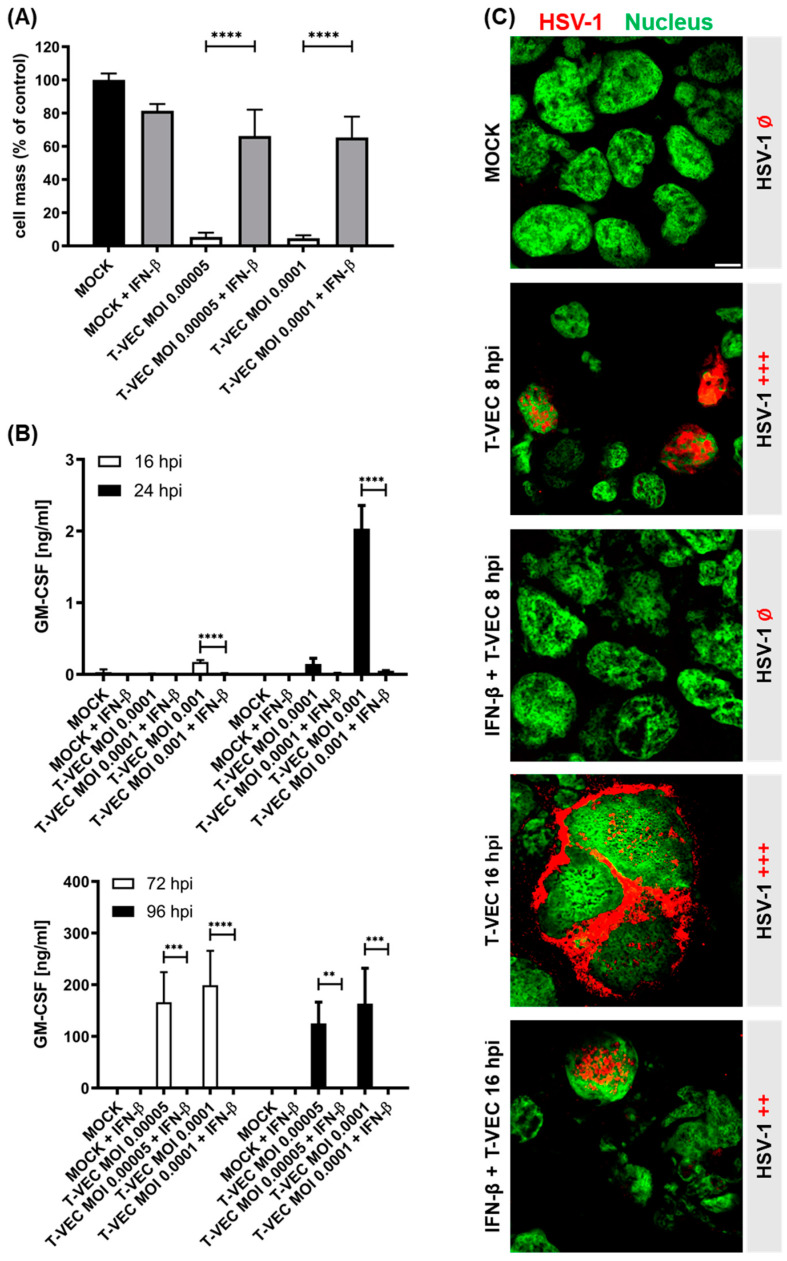
Influence of IFN-β pretreatment on the oncolytic efficacy of T-VEC in 143100 NC cells: (**A**) 143100 NC cells were pretreated with 2 ng/mL IFN-β for 16 h before infection with T-VEC at two different tumor cell line-adjusted multiplicities of infection (MOIs) or remained uninfected (MOCK + IFN-β). At 96 h post infection (hpi), the remaining tumor cells were determined by SRB viability assays. The anti-tumor effect of each treatment modality was calculated relative to untreated control (MOCK). The mean ± SD of at least two independent experiments performed in triplicates is shown. ** *p* < 0.01, *** *p* < 0.001, **** *p* < 0.0001. (**B**) 143100 NC cells were pretreated with IFN-β analogue to (**A**) and infected with T-VEC at MOIs 0.0001 and 0.001 for the timepoints 16 and 24 hpi and at MOIs 0.00005 and 0.0001 for the timepoints 72 and 96 hpi, respectively. At 16, 24, 72, and 96 hpi supernatants were harvested and T-VEC-mediated expression of the marker protein GM-CSF was measured via ELISA. (**C**) Immunofluorescence images of 143100 NC cells pretreated with IFN-β and infected with T-VEC (MOI 0.1) for 8 or 16 h. ++, moderate HSV-1 staining; +++, strong HSV-1 staining; Ø, no HSV-1 staining. Scale bar shows 5 µm.

**Figure 4 cancers-14-02761-f004:**
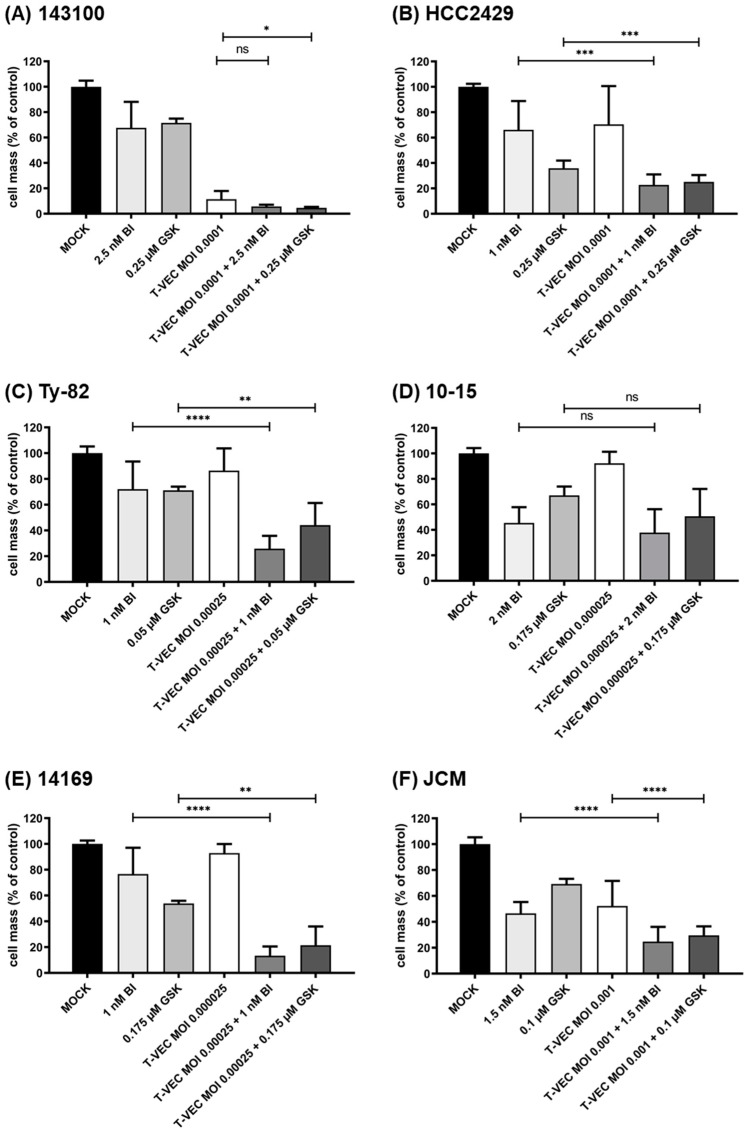
Viability of human NC cell lines after combinatorial treatment with T-VEC and either BET inhibitor (iBET) BI894999 (BI) or GSK525762 (GSK): 143100 (**A**), HCC2429 (**B**), Ty-82 (**C**), 10-15 (**D**), 14169 (**E**), and JCM (**F**) NC cells were treated with T-VEC, BI, and GSK at indicated multiplicities of infection (MOIs) or concentrations alone or with the appropriate combinations or remained untreated (MOCK). MOIs and iBET concentrations were adjusted to the appropriate cell lines. The remaining tumor cells were determined by SRB viability assay at 96 h post infection (hpi). The anti-tumor effect of each treatment modality is calculated relative to MOCK control. The mean ± SD of at least two independent experiments performed in triplicates is shown. Calculated significances always refer to the most potent monotherapy. * *p* < 0.05; ** *p* < 0.01, *** *p* < 0.001, **** *p* < 0.0001, n.s. not significant.

**Figure 5 cancers-14-02761-f005:**
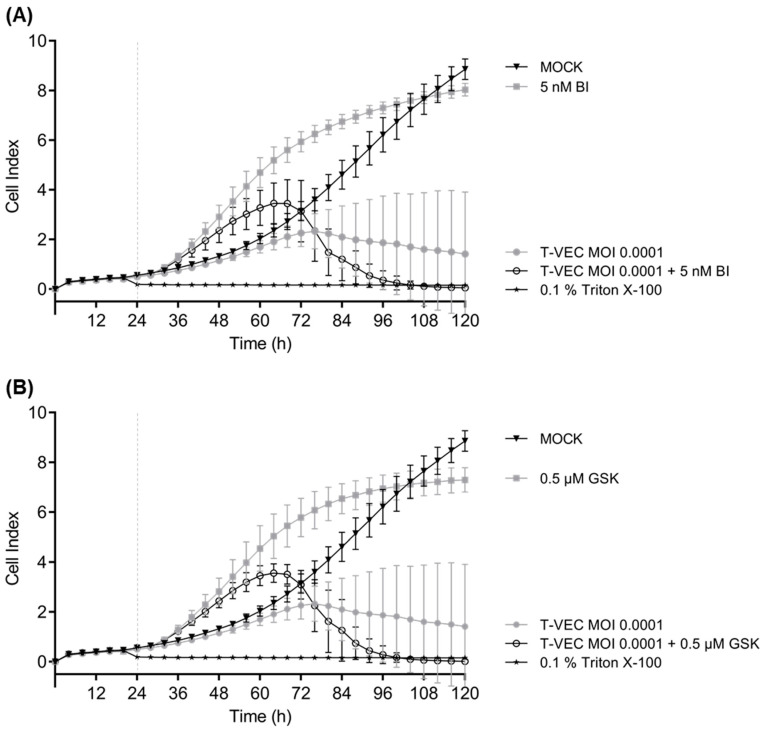
Real-time analysis of 143100 NC cells treated with T-VEC alone or after combinatorial treatment with the BET inhibitors BI894999 (BI) (**A**) or GSK525762 (GSK) (**B**). 24 h after seeding, 143100 NC cells were infected with T-VEC (MOI 0.0001) alone or in combination with 5 nM BI (**A**) or 0.5 µM GSK (**B**) or remained untreated (MOCK). Triton X-100 was added as a negative control inducing maximum lysis of tumor cells. Real-time cell proliferation was monitored using the xCELLigence^®^ RTCA SP system. Measured electrode impedance is expressed as cell index. One representative of two independent experiments performed in triplicates is shown. Vertical dashed lines indicate the time point of T-VEC infection.

**Figure 6 cancers-14-02761-f006:**
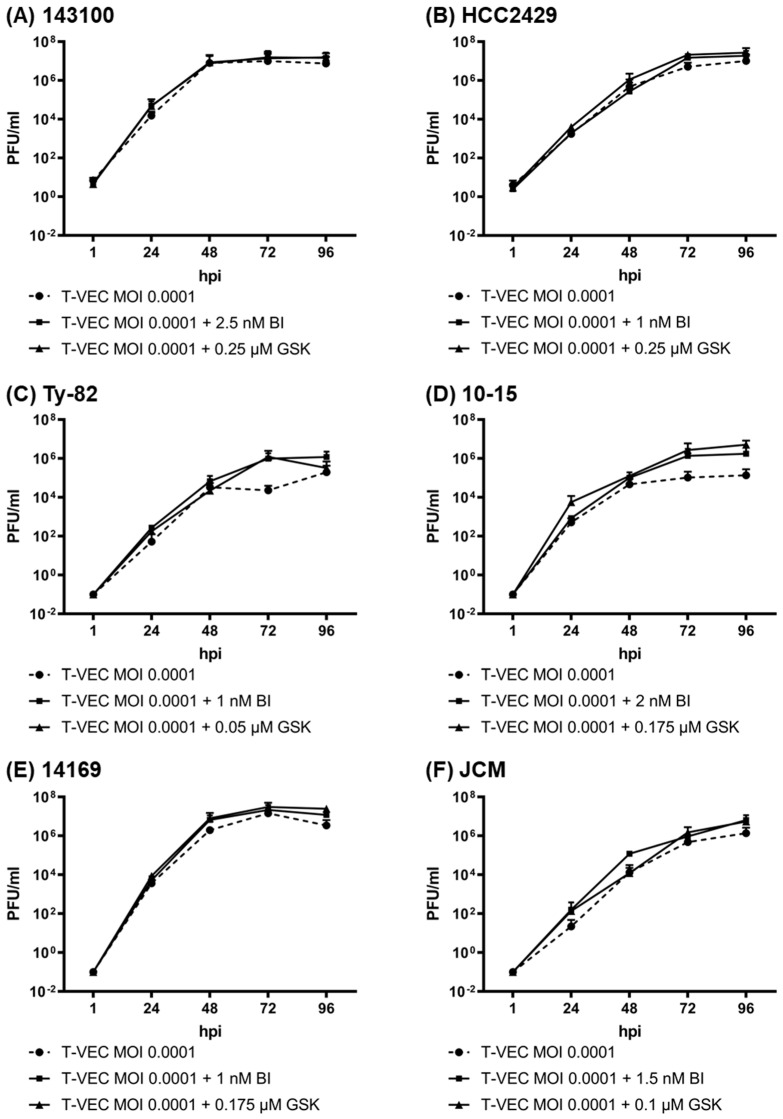
Viral replication of T-VEC in NC cell lines alone and after combination with either BET inhibitor (iBET) BI894999 (BI) or GSK525762 (GSK): 143100 (**A**), HCC2429 (**B**), Ty-82 (**C**), 10-15 (**D**), 14169 (**E**), and JCM (**F**) tumor cells were infected with T-VEC at a multiplicity of infection (MOI) of 0.0001 alone or in combination with BI or GSK at cell line-adjusted concentrations. Viral replication was analyzed via plaque assay at 1, 24, 48, 72, and 96 h post infection (hpi).

## Data Availability

The data presented in this study are available within this article and its Appendix A. Further information is available on request from the corresponding author.

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
