# Peer review of "Efficacy of Oncolytic Herpes Simplex Virus T-VEC Combined with BET Inhibitors as an Innovative Therapy Approach for NUT Carcinoma"

_cancers, 2022, doi:10.3390/cancers14112761_

Round 1
Reviewer 1 Report
In this paper, the authors present data that is mainly convincing regarding the combination therapy of a licensed oncolytic virus, T-VEC with chromatin modifying compounds, iBET inhibitors in killing or inducing a static growth state in cells of a particular cancer, NUT. This is a cell-based study that examines some immune parameters relevant to immuno-virotherapy, such as response to interferon treatment although some examinations of cellular immune responses towards the T-VEC/iBET treated cells would have been interesting.
The Introduction seems to pre-suppose a knowledge of the cancer cell system. Being a rare cancer, definition of terms such as NUT and BET would have been helpful to the general reader.
The Results are logically presented and the Discussion is reasonable if possibly somewhat lengthy.
Reviewer 2 Report
Ohnesorge et al. studies the combination of T-Vec with BET inhibitors(iBET) on NUC carcinoma treatment. T-VEC could efficiently infect and replicate in NC cell lines and showed strong cytotoxic effects. The anti-tumor effect could be enhanced by iBET treatment following viral infection. Viral replication was not impaired by iBET treatment. There are still some concerns of the manuscript that can be improved for consideration of publishing in Cancers journal.
Major point:
- Recommend providing in vivo combinations as the experiments performed are only in vitro.
Minor points:
- Line 29: recommend changing "extremely aggressively growing tumor" to "extremely aggressive tumor". The original phrasing is awkward phrasing.
- Line 60: recommend changing "radiochemotherapy" to "chemoradiation" as this is how it is commonly referred to in the literature.
- Line 354: recommend changing "more rapid slowdown" to "a more rapid time to plateau".
- Line 472: The terminology "partially unsustainable" is vague. Recommend clarifying this statement with more concrete terms.
Author Response
Author's Reply to the Review Report (Reviewer 2)
Comments and Suggestions for Authors
Ohnesorge et al. studies the combination of T-VEC with BET inhibitors (iBET) on NUT
carcinoma (NC) treatment. T-VEC could efficiently infect and replicate in NC cell lines and
showed strong cytotoxic effects. The anti-tumor effect could be enhanced by iBET treatment
following viral infection. Viral replication was not impaired by iBET treatment. There are still
some concerns of the manuscript that can be improved for consideration of publishing in
Cancers journal.
Major point:
1. It is recommended to provide in vivo combinations as the experiments performed are in
vitro only.
Our response:
We appreciate this note saying that cell-based experimental results can be very different from
those from the subsequent animal model-based experiments, especially for developing
immune-related therapies.
However, at this stage of development of our novel approach combining virotherapy and iBET
therapy, there are unfortunately no suitable immunocompetent mouse models available, as
research on this rare cancer so far is not that advanced.
Of note, NC xenograft mouse models do exist, but would not enable an adequate investigation
of the immunological processes involved in this combinatorial immunovirotherapy.
We have already included a corresponding paragraph in the discussion section to explain this
issue to the readership:
Page 18, line 510 - 514: “In a next step, it would be beneficial to test the efficacy of this
therapeutic regimen in NC organoid models and further in immunocompetent mouse models.
However, to date only immunodeficient mouse models could have been developed
[12,37,41,42], which are not suitable to study the immunologic anti-tumor processes triggered
by immunovirotherapy.”
---------------------
Minor points:
1. Line 29: recommend changing "extremely aggressively growing tumor" to "extremely
aggressive tumor". The original phrasing is awkward phrasing.
Our response:
We apologize for this misleading phrase. We have now reworded these lines in the revised
version of our manuscript:
Page 1, Line 29 - 30: “NUT carcinoma (NC) is an extremely aggressive tumor and current
treatment regimens offer patients a median survival of six months only.”
---------------------
2. Line 60: recommend changing "radiochemotherapy" to "chemoradiation" as this is how it
is commonly referred to in the literature.
Our response:
Thank you very much for this correction. In the revised version of our manuscript we now have
used the correct term "chemoradiation".
---------------------
3. Line 354: recommend changing "more rapid slowdown" to "a more rapid time to plateau".
Our response:
Thank you very much. In the revised version of our manuscript we now have employed the
phrase "a more rapid time to plateau".
---------------------
4. Line 472: The terminology "partially unsustainable" is vague. Recommend clarifying this
statement with more concrete terms.
Our response:
Thank you very much for this note. We have now reworded these lines in the revised version
of our manuscript as follows:
Former sentence:
The initial objective of achieving a viable NC tumor cell mass of approximately 75 % with
monotherapeutic treatment was partially unsustainable.
Page 16, Line 480 - 483 (new sentence):
The initial goal of achieving a viable NC tumor cell mass of approximately 75 % with
monotherapeutic treatment was met only with difficulty, as T-VEC proved to be exceedingly
effective in terms of its virus-induced oncolysis of NC cells

Reviewer 3 Report
The authors are presenting a comprehensive study regarding the efficiency of immunotherapy with T-VEC, as monotherapy for treatment of NUT carcinoma cells.
The authors have used a panel of six human NUT carcinoma (NC) cell lines which were infected with the oncolytic herpes simplex virus type-1 (HSV-1) construct T-VEC (Talimogene laherparepvec). The results were evaluated by multiple assays: Sulforhodamine B cell viability assay, Real time cell proliferation assay, Virus growth curve, GM-CSF ELISA, Transmission electron microscopy, Immunofluorescence staining and confocal microscopy.
All NC cell lines were found to be susceptible to viro-therapy, even at very low concentrations of T-VEC. The results are sustained by 6 figures.
Like a continuation of this study, the authors intend to test the efficacy of this therapeutic regimen in NC organoid models and further in immunocompetent mouse models.
The study underlines the benefit of oncolytic viral therapy of tumors.
Round 2
Reviewer 2 Report
N/A